# Potential Infection Risks of the Wheat Stripe Rust and Stem Rust Pathogens on Barberry in Asia and Southeastern Europe

**DOI:** 10.3390/plants10050957

**Published:** 2021-05-11

**Authors:** Parimal Sinha, Xianming Chen

**Affiliations:** 1ICAR-Indian Agricultural Research Institute, New Delhi 110012, India; sinhapath@gmail.com; 2Department of Plant Pathology, Washington State University, Pullman, WA 99164-6430, USA; 3US Department of Agriculture—Agricultural Research Service, Wheat Health, Genetics, and Quality Research Unit, Pullman, WA 99164-6430, USA

**Keywords:** alternate host, *Berberis* spp., *Puccinia striiformis* f. sp. *tritici*, *Puccinia graminis* f. sp. *tritici*, potential infection risk

## Abstract

Barberry (*Berberis* spp.) is an alternate host for both the stripe rust pathogen, *Puccinia striiformis* f. sp. *tritici* (*Pst*), and the stem rust pathogen, *P. graminis* f. sp. *tritici* (*Pgt*), infecting wheat. Infection risk was assessed to determine whether barberry could be infected by either of the pathogens in Asia and Southeastern Europe, known for recurring epidemics on wheat and the presence of barberry habitats. For assessing infection risk, mechanistic infection models were used to calculate infection indices for both pathogens on barberry following a modeling framework. In East Asia, Bhutan, China, and Nepal were found to have low risks of barberry infection by *Pst* but high risks by *Pgt*. In Central Asia, Azerbaijan, Iran, Kazakhstan, southern Russia, and Uzbekistan were identified to have low to high risks of barberry infection for both *Pst* and *Pgt*. In Northwest Asia, risk levels of both pathogens in Turkey and the Republic of Georgia were determined to be high to very high. In Southwest Asia, no or low risk was found. In Southeastern Europe, similar high or very high risks for both pathogens were noted for all countries. The potential risks of barberry infection by *Pst* and/or *Pgt* should provide guidelines for monitoring barberry infections and could be valuable for developing rust management programs in these regions. The framework used in this study may be useful to predict rust infection risk in other regions.

## 1. Introduction

Stripe rust and stem rust, caused by *Puccinia striiformis* f. sp. *tritici* (*Pst*) and *P. graminis* f. sp. *tritici* (*Pgt*), respectively, are among the most important diseases of wheat worldwide, and they pose major threats to global wheat production [1,2,3]. Destructive epidemics can occur over vast areas within few weeks if susceptible cultivars are widely grown and weather conditions are favorable to rust [4,5,6,7]. In wheat, stripe rust has recently emerged as one of the most destructive diseases [2,8,9]. In the last 15 years, the disease has become one of the largest biotic limitations to wheat production and threatens global food supply. In a study on global incidence of wheat rusts over the past 40 years, Morgounov et al. [10] reported significant increases in stripe rust severities between 2001 and 2010 in Central and West Asia. Five devastating stripe rust epidemics have occurred in the Central Asian region since 1999 [11], and the outbreaks were severe, particularly in Uzbekistan [11], Turkey [2], and Iran [12,13]. In 2009 and 2010, severe outbreaks occurred across Central and West Asia and North Africa, suggesting that the disease needs attention at the international level [14].

Given the expanding geographical extent and increased production losses associated with the disease, stripe rust is now the most damaging of all the cereal rusts [2,7,8,9,15,16]. Currently, most of the world’s wheat-producing areas are vulnerable to stripe rust due to the widespread of virulent and aggressive races of *Pst* and lack of adequate resistance in wheat cultivars, and the disease causes an annual global loss of over 5 million tons of wheat with an estimated market value of $1 billion [2,9,17]. Recent changes in the spread of the disease into nontraditional, warmer, and dryer areas suggest a wider range of adaption in the pathogen [17,18]. Changes in cropping systems and climates have been implicated with the recent emergence of stripe rust [9,19]. Stem rust continues to be a potentially devastating disease in wheat and other cereals [20,21]. After years of successful management, the re-emergence of stem rust in Africa and Europe is now an increasing threat to global food security [6,22,23]. Potential annual stem-rust losses have been previously estimated from $7.6 to $53.7 billion globally [17].

Both the stripe rust and stem rust pathogens are macrocyclic, heteroecious fungi with a complete lifecycle consisting of five spore stages. They both have wheat and other cereal crops as their primary hosts and grasses as auxiliary hosts for infection by urediniospores and aeciospores and for producing teliospores and basidiospores. Barberry (*Berberis* spp.) and mahonia (*Mahonia* spp.) serve as their alternate hosts for infection by basidiospores and production of pycniospores and aeciospores [24,25,26,27,28]. Several studies on molecular characterization of *Pst* populations using uredinial samples from wheat suggested sexual recombination occurring in some Asian countries such as China, Nepal, Pakistan, and Turkey [26,29,30,31]. So far, natural infection of barberry by *Pst* has only been found at low frequencies in China [32,33]. However, in Europe and North America, barberry plants have not been found infected by *Pst* but heavily infected by *Pgt* [27,28,34]. In the US Pacific Northwest, *Pgt* basidiospores can infect barberry providing aeciospores to infect and cause stem rust epidemic in wheat and barley. However, *Pst* cannot infect barberry due to teliospore degradation and unmatched barberry phenology [27,28]. Thus, weather factors appear to play an important role in making a difference in barberry infection by the two pathogens. Occurrence of rusts in wheat, other cereal crops, and grasses, and the presence of barberry plants across different geographical regions, especially in Asia and Europe, are well known [24,26,29,35]. The role of barberry for stem rust epidemics has been established for a long time. However, the importance of barberry species as alternate hosts for *Pst* and possible locations or barberry habitats serving for sexual reproduction and their role in disease recurrence are yet to be determined.

Epidemics of rusts are highly dependent upon weather conditions and cropping systems [7,36]. Weather conditions not only affect survival, infection, growth, and asexual reproduction, but also influence different stages throughout the complete sexual cycle, especially teliospore survival, germination, and basidiospore infection on barberry [26,27,37]. However, synchronization between the susceptible stage of alternate hosts and viable teliospores, together with favorable weather conditions, is vital for infection on alternate hosts by basidiospores and subsequent infection on cereals by aeciospores and urediniospores [26,27]. Although the susceptibility of a large number of *Berberis* species has been proven under controlled conditions, natural infection is dependent on several additional factors [26,27]. Firstly, teliospores must germinate at the time when susceptible tissue of alternate hosts is present and must encounter germinating basidiospores under favorable weather conditions. Secondly, susceptible wheat plants must be available within the distance of aeciospore dispersal during the time of aecial formation on alternate hosts. Thirdly, weather conditions must be suitable for the infection on cereals by aeciospores. Therefore, prediction or assessment of basidiospore infection on barberry based on weather conditions is vital to determine the role of alternate hosts in rust epidemics. However, most studies so far have been focused on urediniospore infection without a connection to infection on barberry. For prediction of basidiospore infection, in addition to optimum temperatures, a long duration of leaf wetness (32 h) is an important criterion for *Pst* to infect barberry [27].

The threshold temperatures for both *Pst* and *Pgt* infection and survival are known [7,36,38,39,40,41,42,43,44]. For *Pst* infection on barberry, a minimum of 32 h leaf wetness under the optimum temperature around 10 °C is required for the process of teliospore germination, basidiospore production and germination, and germ tube penetration of barberry leaves [27]. Hourly leaf wetness or a high relative humidity (RH > 95%) period is an important input for the estimation of infection risk dynamics of rusts [36]. Mechanistic models have been used to study plant distributions [45] and the effects of temperature and leaf wetness on infection as a measure of favorable weather [46,47,48,49]. Micro-meteorologically, leaf wetness or high RH periods can be estimated from daily minimum and maximum temperatures and rainfall [47]. Therefore, the rust favorable weather index as an indicator of barberry infection in a specific location or a large-scale region can be assessed since worldwide geospatial weather databases are available.

Infection risk, as a function of temperature and leaf wetness period, is variable in time and space. It is a primary determinant for occurrence and distribution patterns of plant diseases, particularly for rusts. Models or frameworks for assessing infection risks are vital to improve quantitative understanding of large-scale disease dynamics required for the development of strategies, management decisions, research priorities, and the analysis of future scenarios [17,46]. For infection risk assessment, ecological niche models or species distribution models are used to predict disease epidemics in various regions [17,21,50,51,52]. The NAPPFAST internet system has been used for weather-based disease prediction in both time and space [46]. For real-time monitoring of stem rust across wheat-growing regions, a mechanistic modeling framework has been used to provide risk assessments based on the growing season, meteorological suitability for infection, and aerial transport of infectious spores [4,5]. Model assumptions based only on temperature thresholds may be valid for the prediction of urediniospore infection where even a short period of leaf wetness (above 2 h) is enough for the start of infection.

Assessment of infection risk in terms of environmental suitability in possible barberry habitats is necessary for global survey and rust monitoring. With the long history of wheat rusts in Asian and European countries, assessment of environmental suitability for barberry infection for these countries may improve the understanding of rust epidemiology. This can be accomplished by the calculation of a favorable weather index, as a measure of infection risk, especially for the regions where rust epidemics have been known for a long time. The assessment of environmental suitability for basidiospore infection in the presence of barberry is likely to give insights into the role of alternate hosts in rust epidemiology at regional and global scales, which is of great interest for global rust management. The objective of the present study was to assess the suitability and potential infection risk of both *Pst* and *Pgt* on barberry in Asia and the bordering Southeastern European regions using available geospatial meteorological data.

Assessment of environmental suitability for basidiospore infection in 607 locations in these regions was performed based on a rust infection index and a barberry growth index. The infection index as a suitability score and the barberry favorable growth index (thermal times) as an indicator of possible barberry habitats were combined to indicate potential infection risk for comparison among the different regions.

## 2. Results

### 2.1. Stripe Rust Infection Suitability

The infection suitability for *Pst* was estimated as normalized monthly infection index (MINF), and the values for all 607 sites are provided in Appendix A and the numbers and percentages of locations with different scores for the five regions are given in Appendix A. The majority locations in East Asia (EA) and Central Asia (CA) had relatively low MINF values compared with locations in Northwest Asia (NWA) and bordering Southeast European countries (SEC) (Figure 1a). The median values of MINF in EA and CA were very similar, which were slightly higher than that of NWA but lower than that of SEC (Figure 1a). However, the relatively large upper whiskers in CA and NWA compared with the shorter whisker in EA indicated the existence of locations of higher MINF values in CA and NWA than in EA. However, there was one location (Mazraat Matariyet Jbaa, Lebanon) with a high MINF value; countries in SWA had the lowest MINF values. In contrast, SEC had the highest MINF values.

The stripe rust favorable periods measured as Fav-month (MINF > 0) were about 2 to 4 months in EA (particularly in China, Nepal, and Bhutan) and CA (Azerbaijan, Iran, Russia, and Uzbekistan) and 2 to 7 months in NWA, indicated by the upper quartile ranges (Figure 1b). The maximum Fav-month values were up to 10 months in EA and CA and even up to 11 months in NWA (especially Turkey and Georgia). The absence or shorter lower whisker in EA and CA suggested that 25% of the locations had no or short periods favorable for *Pst* to infect barberry. The SEC region had the highest number of locations with high Fav-month values as indicated by the higher position of the box plot.

Few locations in EA (particularly in China, Nepal, and Bhutan) and several locations in CA (Azerbaijan, Iran, Russia, and Uzbekistan), but most of the locations in NWA (Turkey and Georgia) and almost all locations in SEC were noted with high infection index values (MINF > 0.3) along with high Fav-month values (7–11) (Appendix A). Favorable periods for these locations were observed to occur between August to the next June, covering both wheat and barberry growth. The regions in SEC showed favorable conditions almost all year long, except in July and August. Twenty-four locations were noted to have favorable periods, and they were mostly located in the hilly zones around the Caspian Sea and Black Sea.

Infection suitability scores estimated (as the product of the MINF and Fav-month values and the scores assigned) indicated that the proportion of locations with scores suitable for infection increased from the east to west (Figure 1c). In EA, 74.8% of the locations had low to moderate scores (1–2) as compared with CA, with 66.2% of the locations having low to moderate scores (1–2) and 6.1% having high sores (3–4). The proportions of locations with nonzero infection index values were 98.3% and 100% in NWA and SEC, respectively. These data indicate that the probability of finding barberry infected by *Pst* is higher in NWA and SEC than EA and CA (Appendix A).

The environmental suitability index was noted to increase with latitude and decrease with longitude (Appendix A). Association of infection suitability with latitude and longitude indicated their dependence on locations as the likelihood log ratio and Fisher’s exact test were significant (*p* < 0.05). However, the dependence level was weak (low Phi and Cramer’s V values), indicating no strong relationship between infection risk and location feature. The association pattern in terms of latitude and longitude indicated environmental suitability for rust infection is in specific locations represented by a spatial aggregation more towards the NWA region. The proportion of locations with infection suitability was noted to be associated with elevation. The proportion of locations was higher in the range of 500 to 1400 m above sea level than in the locations of higher elevation, and the proportion decreased with increasing elevation (Figure 1d). Only few locations with elevations of 3000 m or above were noted to have suitable conditions for *Pst* to infect barberry.

### 2.2. Stem Rust Infection Suitability

The infection suitability for *Pgt* was estimated as MINF for 607 locations are given in Appendix A, and the numbers and percentages of locations with different scores for the five regions are summarized in Appendix A. The *Pgt* MINF values with a mean of 0.11 across all 607 sites were significantly higher (*p* < 0.0001) than the mean of 0.09 for *Pst* (Appendix A). Distinctly higher *Pgt* MINF values in EA were noted as the median position was higher than those in CA and NWA (Figure 2a). The environmental suitability for *Pgt* as measured by MINF was highest in SEC but lowest in SWA among the five regions. The majority of the locations in SEC had high MINF values as indicated by the elevated level of quartile and median positions.

The favorable period for rust infection estimated as Fav-month for each location is also given in Appendix A. Twenty-four locations were noted to have favorable periods and are mostly in the hilly zones surrounding the Caspian Sea and the Black Sea. The Fav-month values in all these regions occurred between August to following June, covering the growth season for both wheat and barberry. SEC showed favorable conditions almost all year long except July and August. The mean Fav-month value of *Pgt* (0.33) across all 607 locations was significantly higher (*p* < 0.0001) than that of *Pst* (0.25), indicating that the weather conditions in the studied regions are more favorable for *Pgt* than for *Pst* to infect barberry. The median value of Fav-month for SEC was the highest, followed by NWA and EA, and the lowest for SWA (Figure 2b), similar to the general pattern of *Pst* (Figure 1b).

Locations with high infection index values (MINF > 0.3) along with high Fav-month values (7–11) were noted in EA (particularly in Bangladesh, Bhutan, China, and Nepal) and CA (Azerbaijan, Iran, Russia, and Uzbekistan), the majority in NWA (Turkey and Georgia), several in SWA, and almost all in SEC (Appendix A).

The *Pgt* infection suitability pattern was more or less similar to that of *Pst*. Notably, 72.3 and 75.6% of the locations in EA and CA, respectively, had any infection risks but the proportion of locations with high risks was greater in CA (Figure 2c; Appendix A). The NWA and SEC regions had much higher infection risks, with 98.4 and 100% of their locations, respectively, having an infection suitability index of at least L (low score). These data indicate that the probability of finding locations favorable for *Pgt* infection is higher in SEC and NWA than in EA and CA and is the lowest in SWA (24%) (Figure 2c; Appendix A).

The *Pgt* infection index was also observed to be associated with elevation, latitude, and longitude as indicated by the significant likelihood log ratio and Fisher’s exact test (*p* < 0.05). However, the association level was weak (low Phi and Cramer’s V values), indicating some but not strong relationship between rust infection and location features. Similar to *Pst*, the proportion of locations with infection suitability of at least L was noted to be associated with elevation as the frequency of locations was higher in locations of 500 to 1400 m above the sea level, and the proportion decreased with increasing elevation (Figure 2d). Fewer locations with elevations of 3000 m or higher were noted to be suitable for *Pgt*. It was noted that most of the locations suitable for *Pgt* were also suitable for *Pst*, but this was not always the case.

### 2.3. Barberry Growth Index

Barberry growth was measured in terms of monthly thermal index (NHTT). The NHTT value, Fav-month, favorable growth index, growth suitability score, and the number of favorable months for each of the 607 locations are provided in Appendix A, and the number and percentage of locations with various growth suitability scores for each of the five regions are summarized in Appendix A. The NHTT values were similar in EA and CA as indicated by the close median values with wider variations among locations (Figure 3a). The NWA and SEC regions had higher NHTT values than the other regions, especially indicated by their median values. The SWA region had the lowest NHTT value, indicating less suitable conditions for barberry growth. The majority of the locations in all five regions had a period of more than 7 months favorable for barberry growth (Figure 3b).

Suitability for barberry growth was estimated with scores assigned based on the product of NHTT and Fav-month. Most locations in all regions were found to be mostly favorable for barberry growth, except SWA where only 24% of locations had low to moderate suitability (Figure 3c, Appendix A). In EA and CA, about 50 to 68% of the locations had low to moderate suitability, and 26% with high to very high suitability scores. About 52 to 59% of the locations in NWA and SEC had high to very high suitability scores. Countries with reported barberry growth included Bhutan, China, Nepal, India, and Pakistan in EA; Afghanistan, Azerbaijan, Iran, Kyrgyzstan, Russia, Tajikistan in CA; Armenia and Turkey in NWA; and Bulgaria, Croatia, Greece, Italy, Montenegro, Slovenia, Romania, and Ukraine in SEC based on various sources.

The association of barberry growth index with elevation was significant (*p* < 0.05). The proportion of locations (frequency) with conditions suitable for barberry growth was the highest in the elevation range ≤1400 m, dropped quickly in the range between 1400 and 2000 m, and continually decreased in locations of >3000 m (Figure 3d).

### 2.4. Sensitivity Analysis of Parameter b

Infection index (INF) was sensitive to parameter *b*, the rate of change in infection index with h (RH h ≥ 95%). The INF values were relatively stabilized between 0.1 and 0.15 for both *Pst* and *Pgt* (Appendix A). Parameter estimation for the *b* value using 0.117 reported for leaf rust was found to be appropriate for assessing infection index values and suitability comparison among locations. The number of locations or their frequency under a given infection suitability score (particularly for ‘0 or no risk’) varied when the upper limit 0.01 was fixed. Similarly, the number of locations for barberry growth was observed to differ in magnitude, particularly for the no suitability ‘0’ category when the upper limit was fixed at 0.01. The infection index range for other scores did not change much as far as the number of locations in different categories was concerned.

### 2.5. Comparison of Environmental Suitability Patterns Predicted from the Framework and Dymex Model

Environmental suitability in terms of rust infection index predicted by the modeling framework indicated a fair degree of correspondence with the growth index predicted by the Dymex model as far as the rust distribution was concerned. The relative levels of *Pst* risk predicted by the framework for western China, the Indo-Pakistan-Nepal-Bhutan subcontinent CA, NWA, and SEC (Appendix A) were consistent with the predictions using the Dymex model (Appendix A). However, the predictions by the two models appeared different for eastern China and CA. The *Pgt* risk patterns predicted by the framework model and the Dymex model were similar throughout the regions studied (Appendix A). The predictions for barberry growth by the two models were similar for most of China, NWA, and SEC, but quite different for part of EA and CA (Appendix A). The Dymex output indicated a sparse distribution of barberry across EA and CA, but the framework model predicted locations with highly favorable conditions for barberry growth in EA (particularly in the Indo–Nepal–Bhutan–southwestern China region) and CA. Quantitative comparison of rust infection suitability for selected locations in the US Pacific Northwest and western China indicated a moderate to a substantial level of correspondence for both *Pst* and *Pgt* (*k* = 0.50–0.68), while for barberry growth the correspondence was slightly lower, but still fair (*k* = 0.40–0.50) (Appendix A).

### 2.6. Potential Risk for Pst and Pgt Infection on Barberry

Potential risks of *Pst* and *Pgt* were determined for each location considering the suitability for infection index and the barberry growth index (Appendix A). For *Pst* infection, potential risk (with scores 1 to 4) was observed for 254 (42%) locations out of the 607 locations studied, including 103 (41%) in EA, 82 (42%) in CA, 24 (41%) in NWA, 1 (2%) in SWA, and 46% in SEC (Table 1). These locations were distributed across 26 countries, including 15 countries (Bhutan, China, and Nepal in EA; Azerbaijan, Kazakhstan, and Russia in CA; Georgia in NWA; Albania, Croatia, Greece, Italy, Montenegro, Romania, Slovenia, and Ukraine in SEC) with more than half of the locations having various levels of potential risk. However, no locations in EA and SWA had high risks (scores 3 and 4). Among the 18 locations with high risks, 7 were located in CA (2 in Azerbaijan, 2 in Iran, and 3 in southern Russia), 6 in NWA (Turkey), and 5 in SEC (1 in Albania, 2 in Croatia, 1 in Italy, and 1 in Montenegro). In general, SEC had the highest risk score (1.30), followed by NWA (0.75), CA (0.58), and EA (0.50), and the lowest was in SWA (0.02) when considering both the number of locations and the level of risks.

For *Pgt* infection, 308 (51%) out of the 607 locations in 28 countries were found to have various levels of potential risks (Table 2). Seventeen of the 28 countries had more than half of the locations with potential risks. Of the 308 locations, 10 had high-risk scores (3 and 4), including 1 in China, 2 in Azerbaijan, 1 in Russia, 1 in Albania, 1 in Bulgaria, 2 in Croatia, 1 in Italy, and 1 in Montenegro. When considering both the number of locations and the level of risks, SEC had the highest risk score (1.34), followed by NWA (0.76), EA (0.64), and CA (0.58), and again the least in SWA (0.08).

The distribution of potential risks of infection on barberry by *Pst* is illustrated in Figure 4 and that by *Pgt* in Figure 5. The EA region was less favorable for *Pst* than the CA and NWA regions. However, for *Pgt*, the EA region was more favorable than the CA region. The SEC region appeared to be most favorable for both *Pst* and *Pgt*, and the SWA region the least favorable for both the rust fungi for infection on barberry. The majority of locations favorable for *Pst* were also favorable for *Pgt*, but not always. As the maps indicated, areas in southwestern China and contiguous hilly regions of Indo–Pak–Nepal–Bhutan in EA, the Iran-Southern Russia–Kazakhstan–Turkmenistan–Uzbekistan region in CA, the areas surrounding the Black Sea and Caspian Sea (including Azerbaijan, Georgia, and Turkey) in NWA, and the adjacent SEC should be ground searched for monitoring barberry infection by *Pst* and/or *Pgt*.

## 3. Discussion

Potential risks of *Pst* and *Pgt* infections in barberry were assessed for Asia and the adjoining Southeastern European countries. In general, we found comparatively low *Pst* risks in EA, low to high risk in CA, high to very high risk in NWA, and moderate to very high risk in SEC, and a more or less similar risk pattern for *Pgt* across the regions except for some different risk level increments. In EA, particularly Bhutan, China, and Nepal were noted with locations of high potential risks for *Pgt* compared with the relatively low risk for *Pst*. This is the first study of the potential risk of barberry infection by both *Pst* and *Pgt* in a large geographical scale covering the entire Asian region as well as the adjacent European countries.

Environmental suitability for barberry infection by *Pst* and *Pgt* is useful for the prediction of rust outbreaks and distributions, especially for regions where the uredinial stage cannot survive the hot summer and/or cold winter and aeciospores from barberry are essential to initial inocula. The modeling framework used in this study estimates infection risk for the entire process from teliospore germination, through basidiospore production and germination, to penetration of barberry leaves by germ tubes grown from basidiospores as the infection by *Pst* or *Pgt* is strongly constrained by leaf wetness or high RH hours and temperatures [27,36,38,53]. The assessment framework assumes the presence of viable teliospores in the nearby barberry habitats having susceptible tender leaves, stems, and berries. Prediction of rust risk is more realistic when also taking barberry growth into consideration. However, variation in rust resistance, growth stages, and phenology of *Berberis* species were not considered in the present study, which could be considered in future studies focusing on individual regions with clear information of *Berberis* species related to the issues.

The current finding was based on the estimates of favorable infection index (MINF) and period of occurrence used as a measure of infection suitability. The environmental suitability should explain at least a part of the infection process (basidiospore germination and subsequent penetration) involved in the epidemic development. Rust infection suitability and barberry favorable scores are important components to assess possible role of alternate hosts in a particular region and the global in general. Occurrence of stripe rust and stem rust on wheat, other cereals, and grasses and presence of barberry hosts in the geographical regions of the present study is well known [26,32,40,54,55]. Therefore, the assessment of suitability for these components is verifiable.

The mechanistic model used for estimating environmental suitability or infection index is structurally sound, and it explains the infection process coupling the effect of temperature and leaf wetness hours. Rust infection is best explained by incorporating the effects of both temperature and leaf wetness factors [36,38,46,48]. The model prediction corresponded with some real observations in the locations where barberry infection is known [24,26]. Therefore, the model assumptions were valid and thus reasonable to be used for assessing potential infection risk. The current model framework was workable, but its computation was intensive, particularly when daily data was used as input. Available computational tools were used to implement such prediction scheme for the assessment of leaf wetness hours and infection index. The Dymex model predicts mainly the temperature effect on infection using soil moisture in a deterministic setup. It can be useful only when risk estimation is not affected by leaf wetness hours or any moisture parameters. Such model may be useful for predicting urediniospore infection on wheat, for which even 2- or 3-h of high RH on wheat leaves may allow urediniospore infection [4,5,21,37]. However, barberry infection cannot be explained without consideration of a much longer continual duration of high RH or dew formation as the duration needs to be long enough for starting with teliospore germination to basidiospore germination and penetration on barberry leaves [27,37].

Sensitivity analysis of the core parameter *b*, rate of change in infection index with leaf wetness hour, was observed to be stabilized between 0.1 and 0.15 for both *Pst* and *Pgt*. Therefore, the use of *b*-value 0.117, which was reported for leaf rust [48], in the model for estimation of *Pst* and *Pgt* infection indices is justifiable for comparison of infection index values among locations. Potential infection risks for both *Pst* and *Pgt* in China identified in the present study complements the earlier reports of natural infections and the possibility of China being a hotspot for sexual reproduction [22,26,32,33,56]. In China, through artificial inoculation of barberry plants with germinated *Pst* teliospores, 28 *Berberis* species have been identified to be susceptible to the pathogen [32,33]. However, *Pst* aecia have been identified in very low frequencies collected from naturally infected plants of five species (*B. aggregata*, *B. brachypoda*, *B. polyantha*, *B. shensiana*, and *B. soulieana*). The present finding of low to moderate infection risk for *Pst* noted in China is consistent with low *Pst* aecial frequencies on barberry as previously reported [32,33]. High infection risk for *Pgt* in China appears to be well-founded [32]. Low infection potential in the Himalayan hills of Indo–Pak–Nepal–Bhutan reasserts the view of sexual reproduction in the area but with small likelihood. It is possible that barberry infection is occurring but remaining unnoticed due to low infection intensity. Susceptibility of the barberry species collected from Himalayan hills (the Pakistan-Gilgit area) and through artificial infection with teliospores from wheat and infection of wheat with aeciospores reaffirm the view of possible sexual reproduction [30,57]. In Nepal, stripe rust infection in wheat fields during the 2019 crop season and aecia observed on nearby barberry leaves indicated the likely infection risk of stripe rust (https://www.wheatrust.org, accessed on 2 October 2020). The extended Himalayan region, including India, Pakistan, Nepal, and China in EA and the countries in CA and NWA have been inferred as the regions for sexual reproduction [2,12,17,30,58,59,60,61].

The high potential risks for both *Pst* and *Pgt* estimated in the present study indicate that CA NWA, and along with SEC, are the possible regions suitable for both pathogens. In CA and NWA, the large number of locations with high potential risks, particularly in Iran, Azerbaijan, Uzbekistan, southern Russia, Turkey, and Georgia, suggest existence of environmental conditions suitable for both the rust fungi. Stripe rust and stem rust on wheat commonly occur in these countries [11,13,29,54,62,63]. Barberry leaves with aecia were observed in Azerbaijan, Georgia, Iran, Tajikistan, Turkey, and Uzbekistan [13,20,24,64,65], indicating at least one of the rust fungi infecting barberry. The barberry suitability scores in these countries support the high potential of barberry infection. Races of *Pgt* with high variability to *Sr* resistance genes have been identified from aecial samples from the infected barberry plants in Kelardasht and the mountainous areas surrounding Caspian Sea, indicating that barberry species serve as alternate hosts for *Pgt* in these regions [13]. The large number of favorable locations in CA, NWA, and the Balkan region in southeastern Europe identified in the present study surround Caspian, Black, Aegean, and Ionian seas. Therefore, the areas along the coasts of these seas most likely satisfy the conditions for such a prolonged period of leaf wetness (32 h) required for *Pst* teliospore germination and infection. Ground surveys should be conducted in these areas to prove the hypothesis.

The similar geographical distribution patterns for stripe rust and stem rust of wheat and potential barberry infection support the view that the locations favorable for the wheat rusts are also suitable for infection on the alternate hosts. The fair level of barberry suitability noted in the present study predicates the authenticity of these locations for possible barberry infection. Our results of risk prediction based on weather conditions support previous reports as regions of perpetual risk of *Pgt* for barberry infection in the same specified locations of China and Northwestern Asian region have been predicted based on the components of the epidemic process (susceptible wheat crop, viable spore, and weather factors) [4,5,21]. Regions normally having rust infection in wheat crops assure availability of matured teliospores that is the first requirement for barberry infection. Potential infection risk refers to specific conditions for the production and survival of teliospores as well as barberry growth. The majority of the locations that are observed to be common for rust epidemics (urediniospore infection) as well as for rust suitability risk for basidiospore infection on alternate hosts indicate that regions favorable for large-scale urediniospore infection may be the probable regions also for basidiospores infection on barberry [4,5,17,21]. Therefore, the potential risks of both *Pst* and *Pgt* in the locations/regions predicted in the present study is well justifiable.

For estimating the barberry growth index, model parameterization was done based on the growth features of *B. vulgaris* [66]. Therefore, infection suitability described in the present study is with reference to *B. vulgaris*. The suitability may be extended to other species with a similar growth pattern with *B. vulgaris* but may not be applicable to species with very different growth patterns. Bhutan and China in EA and Afghanistan, Azerbaijan, Iran, Kyrgyzstan, Russia, Tajikistan, and Turkmenistan in CA were found to be most favorable for barberry growth throughout the year, which were consistent with previous reports [66]. Our predictions indicated that the Indo–Pak–Nepal–Bhutan–China region had better barberry growth than CA countries. However, the geographical distribution of *B. vulgaris* reported by the Global Biodiversity Facility (https://gbif.org, accessed on 5 December 2020) shows that the Indo-China region has low growth of *B. vulgaris*, but high abundance is shown when all *Berberis* spp. are considered. This is also demonstrated by ground surveys in China, and most of the artificially tested *Berberis* species in China have been reported susceptible to *Pst* infection [32,33,67]. Different *Berberis* species have been reported in India but not including *B. vulgaris* [68]. The barberry growth predicted for different regions in the present study, together with reports in the literature, indicates the availability of susceptible alternate hosts. Therefore, the predicted low to moderate infection risk for the Indo–Pak–Nepal–Bhutan–China region may be ascribed to varying degrees of environmental suitability as well as uneven distribution of barberry both in terms of species and phenology. The rust infection risks estimated in the present study indicate weather conditions in many locations are favorable to either *Pst* or *Pgt* infections during wheat growth ([4,5]; https://gbif.org, accessed on 5 December 2020).

However, the ability to infect barberry is not only dependent on environmental conditions but also dependent of the characteristics of the rust pathogen. Barberry infection in the U.S. Pacific Northwest is important for stem rust epidemics on wheat, but not for stripe rust. Aeciospores of *Pgt* from barberry are essential inoculum for infecting wheat crops because the uredinial stage is unlikely to survive the winter in the region [27,28]. The dormancy of *Pgt* teliospores allows survival during the wet winter with fluctuating temperatures. The wet and fluctuating temperature conditions break the dormancy and allow teliospores to germinate and produce basidiospores at the right time to infect young leaves of barberry in the spring [27]. In contrast, *Pst* telia are mostly buried into the leaf tissue, and teliospores do not have dormancy. Moreover, *Pst* teliospores need a much longer (minimum of 32 h) continual dew-forming conditions to produce basidiospores and infect barberry than *Pgt* or *Pst* urediniospore infection on wheat.

Therefore, a lack of such prolonged period of leaf wetness conditions during the season of telial maturity effectively negates *Pst* infection in the region. The covered telia, lack of teliospore dominancy, requirement of a long period of dew formation, and sensitivity to degradation make *Pst* a poor pathogen to coincide with the susceptible stage of alternate hosts [27]. These genetic and physiological characteristics support the poor link between the *Pst* alternate host and stripe rust epidemics on cereal crops. Based on their results, Wang and Chen [27] pointed out that for being able to infect barberry in a region, *Pst* requires a relative dry winter to allow teliospores to survive and a wet period in spring when barberry plants produce new leaves. Such climate conditions may occur but likely not every year or in every area. Thus, yearly variations in climatic conditions may affect infection and possibly creates a narrow window for *Pst* to infect barberry. In contrast, the coincidence of teliospore germination synchronized with new leaf growth of barberry makes *Pgt* a much better pathogen to infect alternate hosts. Therefore, barberry infection by *Pgt* is more common, and the locations predicted to have high risks can be proven more easily than *Pst*.

Changing the interval range of MINF for no risk or the low-risk category was observed to be sensitive for substantial changes in proportion of risk locations. The number of locations or frequency under infection suitability scores, particularly for ‘0’ or ‘no risk’, varied with the upper limit fixed at 0.01. Similarly, the number of locations for barberry growth was observed to differ in magnitude particularly for the no suitability ‘0’ category, when the upper limit was fixed at 0.01. Estimation of infection suitability can be improved with modification of the interval particularly for the no risk category. More information on barberry infection sites is required. However, changes in the interval upper limit did not affect the comparison of the predicted relative infection risk patterns among the regions. The region-wise risk patterns, instead of individual countries, were more reasonable because the mean values were derived from a large number of sites (>30 sites) and the value distributions are closer to normal. The inclusion of Azerbaijan in NWA because of its closeness to Georgia and Turkey would have given a better representation of the regional risk. However, the overall increase in risk gradient from EA to NWA and SEC across the studied regions remained the same.

The role of alternate hosts includes allowing the pathogen to survive seasons without primary hosts under conditions unfavorable to its uredinial stage, providing initial inoculum to start epidemics on cereal crops, and producing new races or genotypes to diversify pathogen populations [26]. However, infection of alternate hosts by a rust fungus may not occur in all areas where the disease occurs on a cereal crop. As mentioned above, barberry does not play a role in stripe rust epidemics in the US Pacific Northwest, while it is essential for stem rust in the region as barberry provides aeciospores as a major initial inoculum for infecting cereal crop [27,28]. Similarly, *Pgt* is commonly found on barberry in some of the European countries but not *Pst* [34]. For many regions, urediniopores are major inocula for epidemics of rusts, especially stripe rust, on cereals. Urediniospores can disperse hundreds or thousands of kilometers to spread the disease [1,2,3,4,5,12,58]. For generating races and diversifying the pathogen population, sexual recombination is a quick process as a large number of races or genotypes can be detected in a single field or generated from a single isolate [24,28,65,68,69]. However, mutation and somatic recombination have been proven to be major mechanisms for generating variation of both *Pst* and *Pgt* [35,70,71,72,73,74,75]. As the role of alternate hosts on rusts on cereals, especially stripe rust, is less clear in many regions of the world, the present study attempted to estimate the risk of barberry infection by both *Pst* and *Pgt* in Asia and the surrounding European countries.

The data of infection risk on barberry obtained in the present study should be useful in global surveillance on barberry infection by *Pst* and *Pgt* and in the determination of the epidemiological roles of barberry infection in the development of the rust diseases on cereal crops. Potential risk and spatial patterns are necessary information for developing a prioritized plan for barberry surveillance within a country or geographic region. The assessment framework could be used to assess infection risks for other regions—Africa, the rest of Europe, South America, and North America to identify potential locations for barberry infection and their role in sexual reproduction. Further, the information may serve as guidance for monitoring race changes and deciphering the possible evolutionary mechanisms of pathogen variation. The assessment framework for rust favorable weather conditions and the period of occurrence would help monitor and manage rusts on cereal crops and possible other diseases in the regions. The current study provides only comparative risk assessment among the regions of the studied regions. Nevertheless, the findings may be useful in barberry surveillance programs as well as to test hypotheses in resolving many issues on wheat rust epidemics. The hypothesis put forward particularly for stripe rust infection in barberry is how its infection is constrained by several factors in comparison with stem rust.

Yearly variation in weather, comparatively long hours of leaf wetness, absence of telial dormancy, as well as barberry growth and its susceptibility make a narrow window for infection. Further work is needed to determine the infectivity of rust pathogens on different *Berberis* and *Mahonia* species, risk assessment in different locations (grids) within the 1200 m altitudes using precise weather data, and monitoring barberry infection in nearby wheat fields to resolve the long-standing issue of sexual recombination and the role of alternate hosts.

## 4. Materials and Methods

### 4.1. Selection of Locations

To assess environmental suitability based on meteorological data, a total of 607 locations were selected from 37 countries in East Asia (EA), Central Asia (CA), Northwest Asia (NWA), Southwest Asia (SWA), and adjacent Southeast European countries (SEC). The number of the locations in each country, together with reports of stripe rust and stem rust and the presence of *Berberis* spp., is summarized in Table 1. The elevation, latitude, and longitude of these locations are provided in Appendix A. The locations in individual countries for each sub-region were selected using a stratified sampling method. Stratification of categories was based on elevations: negative to 500, 501–1400, 1401–2000, 2001–3000, and above 3001 m above sea level. To represent a sub-region, at least one location was included based on their relative size and homogeneity.

### 4.2. Framework to Assess Potential Infection Risk

For assessment of rust-favorable weather conditions in each location, a mechanistic infection model, where temperature and leaf wetness or high RH (>95%) h were used to estimate an index for rust infection, was implemented. As described in more detail below, rust-favorable weather conditions in terms of monthly infection index (MINF) and the number of favorable months (Fav-month) were estimated separately for *Pst* and *Pgt* at each location. To determine environmental suitability for barberry infection, a favorable index (MINF × Fav-month) was calculated, and a risk category of basidiospore infection was assigned. Similarly, to locate possible habitats for barberry growth, a development growth index (NHTT) and the number of favorable months (Fav-month) were estimated; and based on the favorable growth index (NHTT x Fav-month), a suitability score for barberry presence was assigned. For each location, the potential infection risk of barberry was assessed based on the favorable index score for infection and barberry growth. The assessment framework is illustrated in Figure 6.

### 4.3. Meteorological Data

Daily maximum (T_max_), minimum (T_min_), dew point (T_dew_) temperatures, and precipitation (mm) from 2010 to 2017 were collected from the National Aeronautics and Space Administration (NASA) website Agroclimatology archive (https://power.larc.nasa.gov, accessed on 10 August 2019). The archive is a repository of weather variables (available in 1° latitude by 1° longitude grid) derived from satellite observations and formatted for input to crop models.

### 4.4. Estimation of Hourly Temperature and High RH (>95%) for Rust Infection

Daily minimum (T_min_) and maximum (T_max_) temperatures at each location were converted into hourly temperatures (T_air-h_) using the standard formula [76,77], which provides a smooth transition from minimum to maximum daily air temperatures.
T_air − h_ = (T_max_ + T_min_)/2 + (T_max_ − T_min_)/2 × cos(0.2618 × (h − TimeVar)(1)
where TimeVar is the h of the day corresponding to the time of occurrence of T_max_. Hourly RH (RH_air-h_) was calculated as:RH_air − h_ = 100 × e_a/_e_s_(2)
where e_a_ and e_s_ are the hourly actual and saturated vapor pressures, respectively [76]. Actual and saturated vapor pressures were calculated as:e_a_ = 0.6108 exp [(17.27 × T_dp − hr_)/(T_dp − hr_ + 273.3)](3)
and
e_s_ = 0.6108 exp [(17.27 × T)/(T + 273.3)](4)
where T_dp − hr_ is the hourly dew point temperature. Hourly dew point temperatures (T_dp − hr_) and site-specific T_dp − hrmax_ [78] were calculated as:T_dp − hr_ = Min (T_air − h_, T_dp − hrmax_)(5)
and
T_dp − hrmax_ = 1.510 + 0.83 × T_avmin_ + 0.17 × R_days_(6)
where T_dp − hrmax_ is the maximum dew point temperature (T_dp − hr_), T_avmin_, the monthly average minimum temperature, and R_days_, the monthly days of rainfall (above 3 mm).

### 4.5. Estimation of Environmental Suitability for Infection

First, rust infection indices (*INF*) for *Pst* and *Pgt* were calculated separately using the temperatures and high RH hours (≥95%) at each location. To relate the effect of temperature, a temperature function ƒ(T) was used [46,79], which utilizes the pathogen’s cardinal temperatures to estimate the shape and response:
ƒ(T) = [(T_upper_ − T_air-h_)/(T_upper_ − T_opt_)] × [(T_air-h_ − T_lower_)/(T_opt_ − T_lower_)]^(T_opt_ − T_lower_)/(T_upper_ − T_opt_)(7)
where ƒ(T) = temperature response to rust infection (scaled between 0 and 1); T_lower_ and T_upper_ are the lower and upper thresholds, respectively for infection beyond which infection is assumed to be zero; and T_opt_ = optimum temperature for infection. For the combined effect of temperature (T_air-h_) and high RH hours (h) on infection index (*INF*), Richards’s function that combines RH-hours (h) with temperature-dependent parameters m and ƒ(T) was used [36,48,49,80]:*INF*(*h*) = ƒ(T) × (1 − EXP(−*b* × (*h* − *m*)))(8)
where *h* = RH hours ≥95%; *m* = minimum RH-h required for barberry infection; ƒ(T) = maximum infection index (as asymptote); and *b* = measure of the rate of change in *INF* with *h*. A value *b* = 0.117 as estimated for the leaf rust pathogen *Puccinia triticina* [48] was applied for both *Pst* and *Pgt*; and m values 32 and 24 h were used for *Pst* and *Pgt*, respectively [27].

For each location (with 8-year data), month-wise average *INF* (*AINF*) and then monthly *AINF* (*MAINF*) were calculated. The *MAINF* values were then normalized [81] and expressed as *MINF_i_* for comparison among locations.
(9)$$\mathrm{AINF}=\sum_{k=1}^{8\;year}\left(INF\right)/8$$
(10)$$\mathrm{MAINF}=\sum_{k=January}^{December}\left(AINF\right)\;/12$$
MINF_i_ = (MAINF_i_ − MAINF_min_)/(MAINF_max_ − MAINF_min_)(11)
where *MINF_i_* is the infection index for the *i*-th location and *MAINF_max_* and *MAINF_min_* are the maximum and minimum values for each location, respectively. Location-wise *Fav-month* was calculated by counting the months having *MINF* > 0 and then expressed as a proportion by dividing the value by 12.

To estimate an infection suitability score for each location, the product of rust infection index (*MINF*) and *Fav-month* was categorized into: (1) no (N) risk when the score was 0–0.0090; low (L) risk between 0.0091 and 0.1000; moderate (M) risk for 0.1001–0.3000; high (H) risk 0.3001–0.7000; and very high (VH) risk above 0.7000. Risk scores were assigned based on the presence of barberry infection of *Pst* and/or *Pgt* reported particularly for China, India, and Pakistan among the Asian countries [24,26]. Infection suitability scores were plotted on global and regional maps using software ArcGis 10.0 (ESRI, Redlands, CA, USA).

### 4.6. Barberry Growth Favorable Index

Barberry growth at each location was estimated based on hourly thermal times (*HTT*) using a nonlinear method [82,83]. For estimation of HTT, cardinal temperatures (*T_b_* = 0 °C, *T_u_* = 30 °C, and *T_opt_* = 22 °C) for seed germination with respect to *B. vulgaris* were used [66,84]. For each location (with 8-year data), *HTT* and the monthly average (*AHTT*) were estimated as:*HTT* = [(*T_h_* − *T_b_*)/(*T_opt_* − *T_b_*)] × [(*T_u_* − *T_h_*)/(*T_u_* − *T_opt_*)](*T_u_* − *T_opt_*)/(*T_opt_* − *T_b_*)(12)
(13)$$\mathrm{AHTT}=\sum_{k=1}^{8}\;\left(HTT\right)\;/8$$
where *T_h_* = hourly temperature, *T_b_* = base temperature, *T_u_* = upper threshold temperature, and *T_opt_* = optimum temperature for barberry growth. Monthly *AHTT* (*MHTT*) was then calculated as the mean of 12 months for each location as:(14)$$\mathrm{MHTT}=\sum_{k=January}^{December}\left(AHTT\right)\;/12$$

The *MHTT* values were normalized [81] and expressed as *NHTT* for comparison of barberry growth among locations.
NHTT_i_ = (MHTT_i_ − MHTT_min_)/(MHTT_max_ − MHTT_min_)(15)
where *NHTT_i_* is the favorable growth index for the *i*-th location and *MHTT_max_* and *MHTT_min_* are the maximum and minimum values for the location, respectively. The number of favorable months (*Fav-month*) for each location was calculated by counting the number of months having *NHTT_i_* > 20. Barberry suitability index values for individual locations were estimated as a product of *NHTT_i_* and *Fav-month* and assigned suitability scores based on the global distribution of barberries from the Global Biodiversity Information Facility (https://www.gbif.org, accessed on 23 October 2020) and the existence of published reports of *Berberis* spp. in the region (Table 3). The categories of the barberry growth suitability score are as follows: no or least favorable (N) when the product value was between 0 and 0.1000, low (L) growth, 0.1001 to 0.3000, moderate (M) growth, 0.3001 to 0.5000, high (H) growth, 0.5000–0.7000, and very high (VH) growth when ≥0.7001.

### 4.7. Analysis for Association of Infection Suitability with Geographic Feature

Pearson’s chi-squared tests were conducted to determine associations of rust infection suitability with location elevation, latitude, and longitude.

### 4.8. Comparison of Rust Infection Suitability with Rust Growth Index (GI)

To validate the rust infection suitability indices obtained from the mechanistic modeling framework (Figure 1), *GI*, as a measure of environmental suitability for *Pst* or *Pgt* infection or barberry growth, was predicted using the Dymex model (Hearne Software, Melbourne, Australia). Their growth indices were estimated using fixed moisture parameters (M0 = 0.2, M1 = 0.8, M2 = 1.2, and M3 = 1.8) and temperature thresholds specific to *Pst* (DV0 = 0, DV1 = 9, DV2 = 11, and DV3 = 25), *Pgt* (DV0 = 3, DV1 = 20, DV2 = 22, and DV3 = 30), and barberry (DV0 = 3, DV1 = 20, DV2 = 22, and DV3 = 30). Stress was not considered in the model. The normalized growth index values for *Pst*, *Pgt*, and barberry growth for the regions of interest were then scaled using the same range as used for the assessment of rust infection suitability and barberry favorable growth indices as presented above.

For testing the reliability of infection suitability scores with growth index predicted by the Dymex model, several known locations of barberry infection by *Pst* and/or *Pgt* were chosen (particularly in the U.S. Pacific Northwest and western China). Location-wise rust infection suitability scores and barberry growth index (barberry favorable) values were compared with the predicted growth index (*GI*) categories of rusts and barberry using Cohen’s Kappa [101]. The agreement between the suitability scores was interpreted following the standard guidelines [102].

For each location, potential infection risks for *Pst* and *Pgt* were assessed separately based on the rust infection suitability and barberry growth index or their suitability scores. Individual locations were categorized into no (N), low (L), moderate (M), high risk (H), or very high (VH) risk based on the concurrence of infection suitability and barberry suitability categories. Locations having either only barberry suitability score or rust infection suitability score, but not both, were categorized to be of no (N) risk. A location having the same category (L, M, H, or VH) for both rust infection and barberry growth, potential risk of the same category was assigned for the location. For a location with a higher infection score category but a lower category for barberry growth, or vice versa, the potential risk was assigned to the lower category for the location.

## 5. Conclusions

In the present study, we used mechanistic models to calculate infection indices for both the wheat stripe rust and stem rust pathogens, following similar modeling frameworks for assessing infection risks on barberry. In East Asia, Bhutan, China, and Nepal were found to have low risks of barberry infection by the stripe rust pathogen but high risks by the stem rust pathogen. In Central Asia, Azerbaijan, Iran, Kazakhstan, southern Russia, and Uzbekistan were identified to have low to high risks of barberry infection for both pathogens. In Northwest Asia, risk levels of both pathogens in Turkey and the Republic of Georgia were determined to be high to very high. In Southwest Asia, no or low risk was found. In Southeastern Europe, similar high or very high risks for both pathogens were noted for all countries. Although the potential risks of barberry infection by the rust fungi were estimated based on chronological weather s, the results should be useful to guide monitoring barberry infections in the risky regions and could be valuable for developing rust management strategies in these regions. The modeling framework used in this study may be useful to predict rust infection risk on cereal crops and alternate hosts in other regions.

## Figures and Tables

**Figure 1 plants-10-00957-f001:**
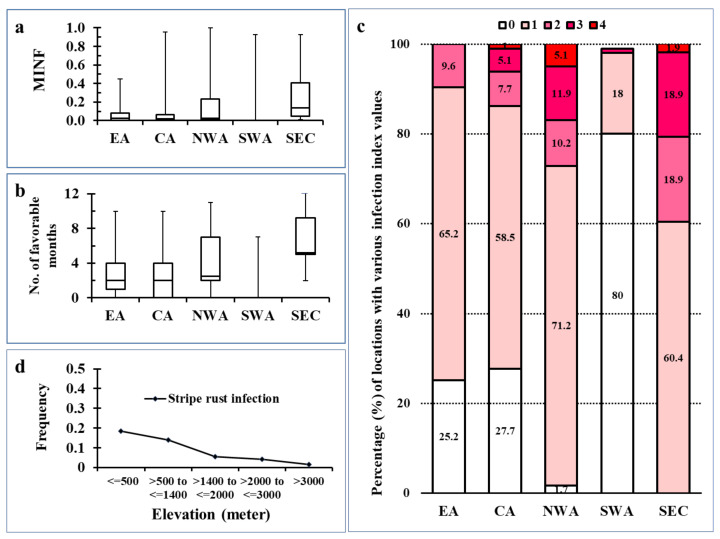
Normalized monthly infection index (MINF), Fav-month, and environmental suitability scores for the stripe rust pathogen (*Puccinia striiformis* f. sp. *tritici*) infection on barberry in Asian and adjacent Southeastern European regions. (**a**): boxplot for MINF values; (**b**): boxplot for numbers of favorable months; (**c**): frequencies of locations in different elevation intervals; and (**d**): percentages of locations with suitability scores in various regions. EA = East Asia; CA = Central Asia; NWA = Northwest Asia; SWA = Southwest Asia; and SEC = Southeast European Countries. Suitability scores: 0 = no; 1 = low; 2 = moderate; 3 = high; and 4 = very high.

**Figure 2 plants-10-00957-f002:**
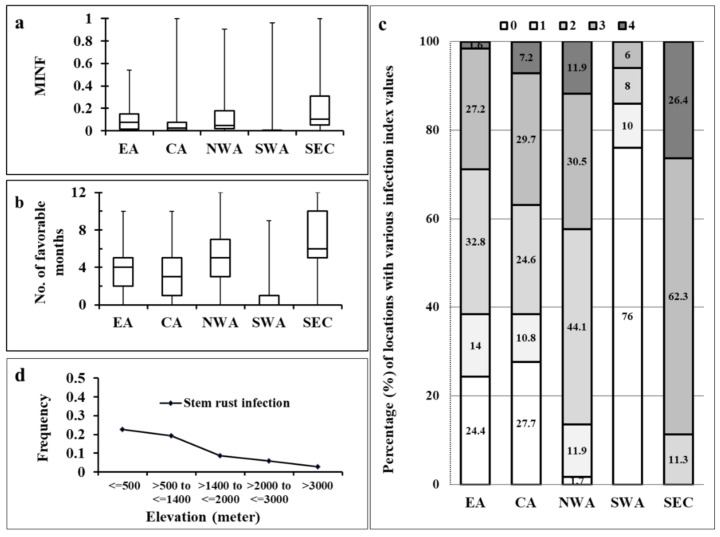
Normalized monthly infection index (MINF), Fav-month, and environmental suitability scores for the stem rust pathogen (*Puccinia graminis* f. sp. *tritici*) infection on barberry in Asian and adjacent Southeastern European regions. (**a**): boxplot for the MINF values; (**b**): boxplot for the numbers of favorable months; (**c**): frequencies of locations in different elevation intervals; and (**d**): percentages of locations with suitability scores in various regions. EA = East Asia; CA = Central Asia; NWA = Northwest Asia; SWA = Southwest Asia; and SEC = Southeast European Countries. Suitability scores: 0 = no; 1 = low; 2 = moderate; 3 = high; and 4 = very high.

**Figure 3 plants-10-00957-f003:**
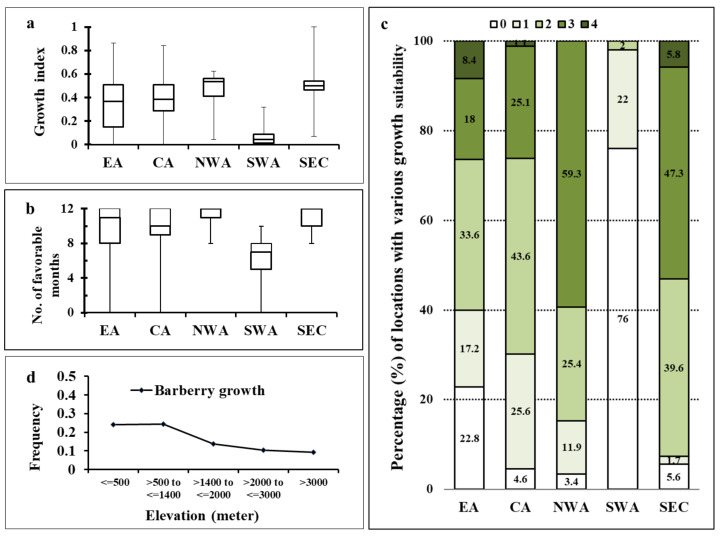
Normalized hourly thermal times (NHTT), Fav-month, and environmental suitability scores for barberry (*Berberis vulgaris*) in Asian and adjacent Southeastern European regions. (**a**): boxplot for NHTT values; (**b**): boxplot for numbers of favorable months; (**c**): frequencies of locations in different elevation class interval; and (**d**): percentages of locations with growth suitability scores in various regions. EA = East Asia; CA = Central Asia; NWA = Northwest Asia; SWA = Southwest Asia; and SEC = Southeast European Countries. Suitability scores: 0 = no or least favorable; 1 = low; 2 = moderate; 3 = high; and 4 = very high growth.

**Figure 4 plants-10-00957-f004:**
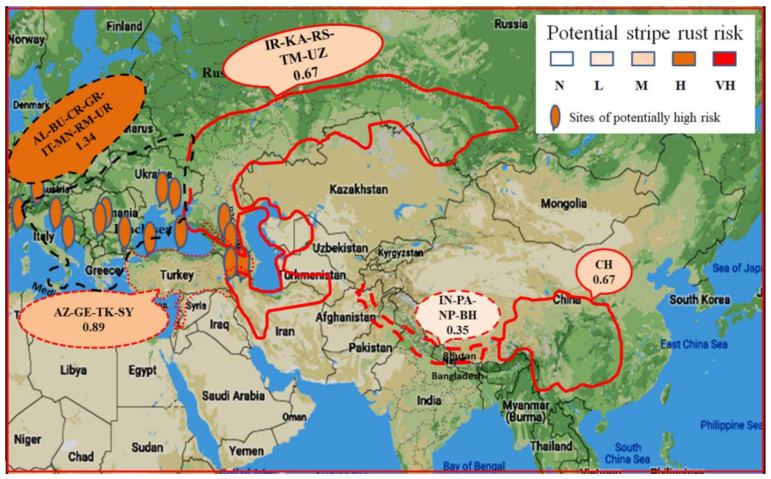
Areas of potential infection risks for the stripe rust pathogen (*Puccinia striiformis* f. sp. *tritici*) in Asia including China, hilly regions of Indo–Pak–Nepal–Bhutan (IN-PA-NP-BH), Central Asia (IR-KA-RS-TM-UZ), and West Asia surrounding Black Sea and Caspian Sea (GE-AZ-TK). AZ = Azerbaijan, BH = Bhutan, CH = China, IN = India, IR = Iran, KA = Kazakhstan, NP = Nepal, PA = Pakistan, RS = Russia, TM = Turkmenistan, and UZ = Uzbekistan.

**Figure 5 plants-10-00957-f005:**
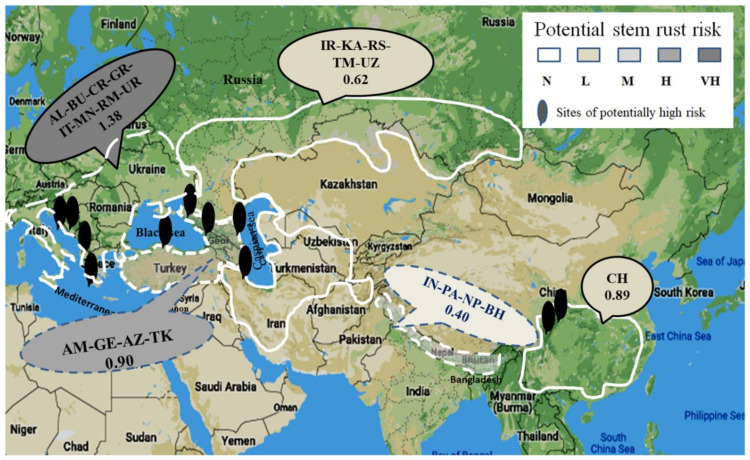
Areas of potential infection risks for the stem rust pathogen (*Puccinia graminis* f. sp. *tritici*) in Asia, including China, hilly regions of Indo–Pak–Nepal–Bhutan (IN-PA-NP-BH), Central Asia (IR-KA-RS-TM-UZ), and West Asia surrounding Black Sea and Caspian Sea (AM-GE-AZ-TK). AM = Armenia, AZ = Azerbaijan, BH = Bhutan, CH = China, IN = India, IR = Iran, KA = Kazakhstan, NP = Nepal, PA = Pakistan, RS = Russia, TM = Turkmenistan, and UZ = Uzbekistan.

**Figure 6 plants-10-00957-f006:**
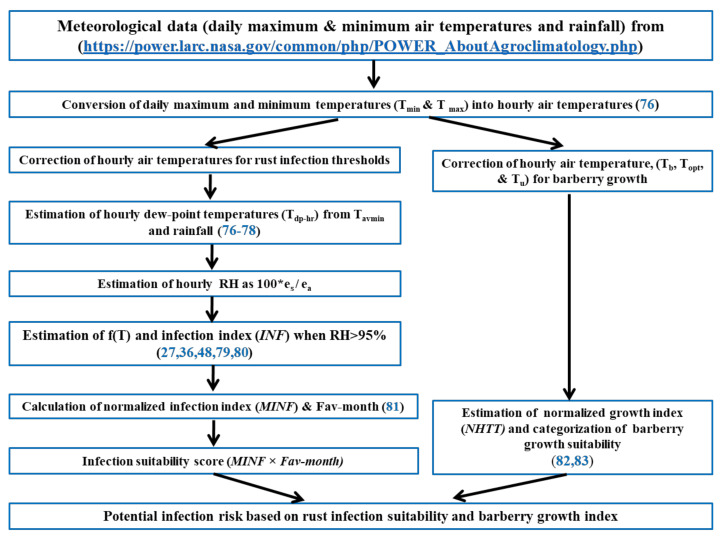
Modeling framework in the assessment of environmental suitability for basidiospore infection in barberry based on meteorological data (https://power.larc.nasa.gov/, accessed on 10 August 2019).

**Table 1 plants-10-00957-t001:** Locations of potential infection risk from *Puccinia striiformis* f. sp. *tritici* in Asian and Southeast European countries.

Region (No. of Locations)	No. of Locations with Risk Scores					Potential Risk	
	1	2	3	4	Total	Proportion ^a^	Risk Score ^b^
*East Asia*							
Bhutan (9)	5				5	0.56	0.56
China (142)	51	22			73	0.51	0.67
India (45)	11				11	0.24	0.24
Nepal (15)	10				10	0.67	0.67
Pakistan (16)	4				4	0.25	0.25
*Regional total (250)*	*81*	*22*			*103*	*0.41*	*0.50*
*Central Asia*							
Afghanistan (20)	1				1	0.05	0.05
Azerbaijan (16)	6	3	1	1	11	0.69	1.19
Iran (58)	11	10	2		23	0.40	0.64
Kazakhstan (18)	9	1			10	0.56	0.61
Kyrgyzstan (14)	2				2	0.14	0.14
Russia (23)	16	1	3		20	0.87	1.17
Tajikistan (13)	3				3	0.23	0.23
Turkmenistan (20)	6				6	0.30	0.30
Uzbekistan (13)	4	2			6	0.46	0.62
*Regional total (195)*	*58*	*17*	*6*	*1*	*82*	*0.42*	*0.58*
*Northwest Asia*							
Georgia (9)	5	2			7	0.78	1.00
Turkey (40)	5	6	6		17	0.43	0.88
*Regional total (59)*	*10*	*8*	*6*		*24*	*0.41*	*0.75*
*Southwest Asia*							
Syria (3)	1				1	0.33	0.33
*Regional total (50)*	*1*				*1*	*0.02*	*0.02*
*Southeast European countries*							
Albania (1)			1		1	1.00	3.00
Bulgaria (15)	4	2			6	0.40	0.53
Croatia (2)			1	1	2	1.00	3.50
Greece (4)	1	3			4	1.00	1.50
Italy (5)	1	3	1		5	1.00	2.00
Montenegro (1)			1		1	1.00	3.00
Romania (11)	10	1			11	1.00	1.09
Slovenia (1)		1			1	1.00	2.00
Ukraine (15)	10	4			14	0.93	1.20
*Regional total (53)*	*26*	*14*	*4*	*1*	*45*	*0.85*	*1.30*

^a^ Total number of locations in all risk categories (1–4) divided by the total number of locations. ^b^ Sum of potential risk scores (no. location × potential risk score) divided by the total number of locations.

**Table 2 plants-10-00957-t002:** Locations of potential infection risk from *Puccinia graminis* f. sp. *tritici* in Asian and Southeast European countries.

Region (No. of Locations)	No. of Locations with Risk Scores					Potential Risk	
	1	2	3	4	Total	Proportion ^a^	Risk Score ^b^
*East Asia*							
Bhutan (9)	3	2			5	0.56	0.78
China (142)	77	23	1		101	0.71	0.89
India (45)	13				13	0.29	0.29
Nepal (15)	9				9	0.60	0.60
Pakistan (16)	5				5	0.31	0.31
*Regional total (250)*	*107*	*25*	*1*		*133*	*0.53*	*0.64*
*Central Asia*							
Afghanistan (20)	1				1	0.05	0.05
Azerbaijan (16)	9	4	2		15	0.94	1.44
Iran (58)	16	9			25	0.43	0.64
Kazakhstan (18)	15				15	0.83	0.83
Kyrgyzstan (14)	1				1	0.07	0.07
Russia (23)	19	3	1		23	1.00	1.22
Tajikistan (13)	1				1	0.08	0.08
Turkmenistan (20)	2	1			3	0.15	0.20
Uzbekistan (13)	5	1			6	0.46	0.54
*Regional total (195)*	*68*	*18*	*3*		*89*	*0.46*	*0.58*
*Northwest Asia*							
Armenia (10)	9				9	0.90	0.90
Georgia (9)	7	1			8	0.89	1.00
Turkey (40)	9	9			18	0.45	0.68
*Regional total (59)*	*25*	*10*			*35*	*0.59*	*0.76*
*Southwest Asia*							
Iraq (5)	1				1	0.20	0.20
Syria (3)	3				3	1.00	1.00
*Regional total (50)*	*4*				*4*	*0.08*	*0.08*
*Southeast European countries*							
Albania (1)			1		1	1.00	3.00
Bulgaria (15)	6	2	1		9	0.60	0.87
Croatia (2)			1	1	2	1.00	3.50
Greece (4)		2			2	0.50	1.00
Italy (5)		4	1		5	1.00	2.20
Montenegro (1)			1		1	1.00	3.00
Romania (11)	11				11	1.00	1.00
Slovenia (1)		1			1	1.00	2.00
Ukraine (15)	11	3			14	0.93	1.13
*Regional total (53)*	*28*	*12*	*5*	*1*	*46*	*0.87*	*1.34*

^a^ Total number of locations in all risk categories (1–4) divided by the total number of locations. ^b^ Sum of potential risk scores (no. location × potential risk score) divided by the total number of locations.

**Table 3 plants-10-00957-t003:** Number of sites in Asian and adjoining Southeast European countries selected for assessment of favorable conditions for barberry infection by *Puccinia striiformis* f. sp. *tritici* and *P. graminis* f. sp. *tritici*, references for the occurrence of stripe rust and stem rust, and presence of *Berberis* spp. and references.

Regions	No. of Locations	References of Stripe and Stem Rusts on Cereals	*Berberis* spp. (References)
*East Asia*			
Bangladesh	5	[49,85]	*Berberis* spp. (GBIF ^a^)
Bhutan	9	[86,87]	*Berberis* spp. [64]; (GBIF)
China	142	[22,88]	*B. aggregata*, *B. aggregata* var. *integrifolia*, *B. atrocarpa*, *B. brachypoda*, *B. chinensis*, *B. circumserrata*, *B. dasystachya*, *B. davidii*, *B. ferdinandi-coburgii*, *B. guizhouensis*, *B. holstii*, *B. jamesiana*, *B. koreana*, *B. phanera*, *B. platyphylla*, *B. poiretii*, *B. potaninii*, *B. shensiana*, *B. soulieana*, *B. stenostachya*, *B. vulgaris*, *B. wangii*, etc. [32,33,89]; (GBIF)
India	45	[87,90]	*B. aristata* [91]; (GBIF)
Mongolia	10	[12,58]	Data not found
Myanmar	8	[55]	*Berberis* spp. [64]; (GBIF)
Nepal	15	[12,87,92]	*B. aristata* [91]; (GBIF)
Pakistan	16	[12,93,94]	*B. aitchinsoni*, *B. baluchistanica*, *B. brandisiana*, *B. brevissima*, *B. calliobotrys*, *B. chitria*, *B. glaucocarpa*, *B. huegeliana*, *B. jaeschkeana*, *B. kashmirana*, *B. kunawurensis*, *B. lyceum*, *B. orthobotrys*, *B. pachyacantha*, *B. parkeriana*, *B. pseudumbellata*, *B. royleana*, *B. stewartiana*, *B. ulicina*, *B. vulgaris* [95]; (GBIF)
*Central Asia*			
Afghanistan	20	[12]	*B. vulgaris* [96]; (GBIF)
Azerbaijan	16	[13,29]	*B. vulgaris* [24,66]; (GBIF)
Iran	58	[16,54]	*B. vulgaris* [34,64,66]; (GBIF)
Kazakhstan	18	[63]	*B. vulgaris* [24,66]; (GBIF)
Kyrgyzstan	14	[12,58]	*B. vulgaris* [24,66];(GBIF)
Russia	23	[16,92]	*Berberis* spp. (GBIF)
Tajikistan	13	[97]	*B. vulgaris* [24,66]; (GBIF)
Turkmenistan	20	[11]	*B. vulgaris* [24,66]; (GBIF)
Uzbekistan	13	[16,97]	*B. vulgaris* [24,66]; (GBIF)
*Northwest Asia*			
Armenia	10	[11,29]	*Berberis* spp. (GBIF)
Georgia	9	[29]	*B. vulgaris* [96]; (GBIF)
Turkey	40	[13,16]	*B. vulgaris* [96]; (GBIF)
*Southwest Asia*			
Egypt	20	[13,93,98,99]	*Berberis* spp. (GBIF)
Iraq	5	[13,16,93]	*Berberis* spp. (GBIF)
Israel	2	[13,93]	*Berberis* spp. (GBIF)
Lebanon	1	[13,93]	*Berberis* spp. (GBIF)
Oman	4	[13,93]	*Berberis* spp. (GBIF)
Saudi Arabia	11	[13,93]	*Berberis* spp. (GBIF)
Syria	3	[13,93]	*Berberis* spp. (GBIF)
Yemen	4	[13,93]	*Berberis* spp. (GBIF)
*Southeast Europe*			
Albania	1	Data not found	*Berberis* spp. (GBIF)
Bulgaria	10	[93]	*Berberis* spp. (GBIF)
Croatia	2	[16]	*Berberis* spp. (GBIF)
Greece	6	[13]	*Berberis* spp. (GBIF)
Italy	5	[13,16]	*Berberis* spp. (GBIF)
Montenegro	1	Data not found	*Berberis* spp. (GBIF)
Romania	12	[62]	*Berberis* spp. [100]; (GBIF)
Slovenia	1	Data not found	*Berberis* spp. (GBIF)
Ukraine	15	[16]	*Berberis* spp. (GBIF)
Total	607		

^a^ GBIF = Global Biodiversity Information Facility (https://www.gbif.org, accessed on 23 October 2020).

## Data Availability

All data used in and created by this study are included in this publication as tables, figures, and Appendix A.

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
