# Peer review of "Potential Infection Risks of the Wheat Stripe Rust and Stem Rust Pathogens on Barberry in Asia and Southeastern Europe"

_plants, 2021, doi:10.3390/plants10050957_

Round 1
Reviewer 1 Report
Infection is an effect of rust race congruence with the host. Even this level of interactions is complicated enough to be nearly unpredictable. Severeness of infection is the result of pathogen population races aggressiveness towards the host. To broken host resistance barriers the rusts evolve gaining new virulences. The level of virulence, and in consequence the disease severity is not only the effect of sexual reproduction taking place on barberry. Mutations taking place during asexual reproduction are in some regions (e.g. Australia) the only source of new races. The worst Puccinia striiformis race, Ug99 has no visible recombination events within chromosomes and no reassortment of chromosomes from different nuclei. These facts are inconsistent with a sexual origin and strongly support that the Ug99 lineage arose by a somatic hybridisation event.
And one more - WIND - the main spores carrier. In some papers was mentioned that on the field infected with rust surrounded by barberry bushes the source of inoculum was external. In our observation the local population structure, complexity and aggressiveness are affected by external spores.
Some minor remarks:
Line 19 etc.
Turkey and Georgia represent West Asia. On the Northwest, we have only Russia.
Line 45
Explain why 88% of the world’s wheat production is susceptible to stripe rust.
Line 56
Check the numbers, please.
Line 75
It should be earlier in this paragraph mentioned about sexual reproduction on the alternate host because, for example, the sentence in lines 66-68 won't be clear for all readers.
Line 97
Duration of leaf wetness - it should be explained here not at line 122.
Line 145 etc.
You need to use the full names, and after that - shortcuts.
Line 242
Have you checked if wheat can be grown on such elevation?
Line 651
..for seed germination with respect to B. vulgaris were used - what does it mean?
Line 700
Potential infection risk. - should be deleted.
Summarizing, please, add information about somatic mutations, about wind influence - 'rust paths' etc. and about rust epidemics in regions without an alternate host - barberry.
Author Response
Dear Reviewer 1:
Thank you for your valuable comments. The attached file lists our responses to all your comments.
Xianming Chen

Reviewer 2 Report
In the present study, the authors investigated the infection risk of wheat stripe rust and stem rust on Barberry (Berberis spp.), an alternate host, in Asia and Southeastern Europe. Authors used different mechanistic models to calculate infection indices using temperature and leaf wetness hours. Authors successfully identified countries/geographical regions with different level of risks for both pathogen. Results of this study will be useful for developing management strategies for the two rusts, particularly for regions under this study.
Overall, the study is well structured. I do not have any special minor or major revision to suggest to the authors and I believe that the study is ready to be published. Great job!
Author Response
Dear Reviewer 2:
Thank you for your positive evaluation. We have improved the manuscript. The attached file lists our responses to your comments.
Xianming Chen
